# Left ventricular reverse remodeling: A predictor of survival in chagasic cardiomyopathy patients with a reduced ejection fraction

**Maria Tereza Sampaio de Sousa Lira**[id]¤*, **Silas Ramos Furquim,
Daniel Catto de Marchi, Pamela Camara Maciel, Rafael Cavalcanti Tourinho Dantas,
Bruno Biselli, Paulo Roberto Chizzola, Robinson Tadeu Munhoz,
Felix José Alvarez Ramires, Barbara Maria Ianni, Fábio Fernandes,
Silvia Moreira Ayub-Ferreira, Eduardo Gomes Lima, Edimar Alcides Bocchi**¤

Instituto do Coração (InCor), Hospital das Clinicas da Faculdade de Medicina da Universidade de São Paulo, Sao Paulo, SP, Brazil

¤ Current Address: Department of Heart Failure, Instituto do Coração (InCor), Hospital das Clinicas da Faculdade de Medicina da Universidade de São Paulo, Sao Paulo, SP, Brazil.
* mtereza.sampaiolira@gmail.com

## Abstract

### Background

Chagas disease is a major health issue in Latin America and is now spreading globally because of migration. Chronic Chagasic cardiomyopathy (CCC) leads to heart failure with a reduced ejection fraction (HFrEF). Left ventricular reverse remodeling (LVRR), defined as an improved LVEF, is associated with improved outcomes in patients with other HFrEF etiologies. Therefore, we evaluated the relationship between LVRR and survival in CCC patients with an LVEF<40%.

### Methods

This retrospective, single-center study included patients diagnosed with CCC and LVEF<40% between January 2006 and September 2021. Patients were divided into two groups: positive RR (PRR; LVEF≥40% or an absolute LVEF increase of ≥10%) and negative RR (NRR). Propensity score matching (PSM) was used to account for baseline differences, and Cox proportional hazards models were applied to determine independent predictors of mortality and heart transplantation.

### Results

A total of 1,043 patients were evaluated; 221 (21.2%) were classified as having PRR, and 822 (78.8%) were classified as having NRR. PRR status was associated with a 55% lower risk of all-cause mortality and heart transplantation over 15 years (p = 0.002). Multivariate Cox analysis revealed that predictors of total mortality and heart transplantation included NRR status, a worse NYHA class, lower serum sodium

**Data availability statement:** The data underlying this study contain sensitive patient information, including personal identifiers such as names, dates of birth, and medical records. Institutional regulations, ethical considerations, and the Brazilian General Data Protection Law (Lei Geral de Proteção de Dados – LGPD, Law No. 13,709/2018) prohibit the public sharing of these data. Access to anonymized data may be provided upon reasonable request, subject to approval by the Institutional Review Board (IRB) of the Heart Institute, University of São Paulo School of Medicine (InCor - FMUSP). Requests can be directed to ccientifica@incor. usp.br. Following publication, data will only be made available upon request and with the approval of the InCor Ethics Committee, ensuring compliance with institutional, ethical, and legal guidelines.

**Funding:** The author(s) received no specific funding for this work.

**Competing interests:** I have read the journal's policy, and the authors of this manuscript have the following competing interests: Maria Tereza Sampaio de Sousa Lira received honoraria from AstraZeneca. Edimar Bocchi received consulting fees from Servier, AstraZeneca, and Boehringer Ingelheim; subsidized travel/ registration/hotel fees from Servier; membership in steering committees from Servier, Novartis, and Boehringer Ingelheim; research grant support through the Heart Institute from Janssen, Bayer/Merck, AstraZeneca, Boehringer Ingelheim, Pfizer, Novartis, Cardiol Therapeutics, and Eurofarma; and honoraria from Servier, Novartis, AstraZeneca, and Boehringer Ingelheim. Eduardo Lima received grants from Novartis, Novo Nordisk, AstraZeneca, and Daiichi Sankyo. Silvia Moreira Ayub-Ferreira received honoraria from Abbott, Novartis, and CSL Vifor. Bruno Biselli received honoraria from Abbott. José Felix Ramires received grants from Novartis. The other authors declare that they have no known competing financial interests or personal relationships that could have appeared to influence the work reported in this paper.

levels, larger LV dimensions, and moderate-to-severe tricuspid regurgitation (TR). The PRR predictors were smaller LV dimensions, less mitral regurgitation, and the absence of triple therapy at baseline. NRR patients were more likely to be on triple therapy at baseline.

## Conclusions

PRR improves survival in CCC patients with HFrEF. Identifying patients with potential for LVRR, alongside early therapeutic interventions, may reduce mortality in this population. Future research should focus on therapies that promote LVRR in patients with CCC.

## Author summary

In this study, we explored the impact of left ventricular reverse remodeling (LVRR) on survival in patients with Chagas cardiomyopathy (CCC) and heart failure with reduced ejection fraction (HFrEF). Chagas disease, caused by the parasite Trypanosoma cruzi, is a major cause of heart failure in Latin America and is increasingly recognized globally due to migration. We found that patients who experienced positive reverse remodeling (PRR), defined as an improvement in heart function, had a 55% lower risk of death or heart transplantation over a 15-year period compared to those with negative reverse remodeling (NRR). Notably, patients with larger left ventricular diameters, moderate to severe mitral regurgitation, and prior initiation of triple therapy were less likely to achieve PRR, suggesting a subgroup of non-responders to standard heart failure treatments. Our findings highlight the importance of early detection and tailored therapies to promote heart recovery in CCC patients. This research provides new insights into managing this challenging condition and underscores the need for further studies to explore novel treatments that could improve outcomes for these patients. By focusing on therapies that enhance heart recovery, we hope to reduce mortality and improve quality of life for individuals living with CCC and HFrEF.

## Introduction

Chagas disease remains a major public health concern in endemic regions such as Latin America, but it has also become an emerging issue in the United States, Asia, and Europe due to human migration [1]. Caused by infection with *Trypanosoma cruzi*, Chagas disease can lead to severe cardiac manifestations, including heart failure (HF) with reduced ejection fraction (HFrEF) [2]. In general, chronic Chagasic cardiomyopathy (CCC) is characterized by a persistent cardiac inflammatory process and the development of fibrosis secondary to the persistence of the parasite in the myocardial tissue [2].

In other non-CCC HFrEF etiologies, left ventricular (LV) injury can be improved or reversed to normal structure and function, either spontaneously or in response to therapeutic interventions. This process is referred to as LV reverse remodeling (LVRR) [3,4] Patients with LVRR have better clinical outcomes, with LVRR cohorts presenting a 5-year survival rate between 80% and 90% and patients who maintain ventricular dysfunction presenting a 65–75% 5-year survival rate [5–7]. Improvement in LVEF is independently associated with lower rates of mortality, hospitalizations, and heart transplantation [8,9].

Only one previous study has evaluated LVRR in patients with CCC [10]. However, there was no significant reduction in mortality associated with the occurrence of LVRR [10]. Therefore, the main objective of this study was to assess whether LVRR is associated with improved survival in patients with CCC and an LVEF<40%.

## Methods

### Ethics statement

This study was approved by the Ethics Committee for Analysis of Research Projects of the Clinical Board of the Instituto do Coração (InCor), Hospital das Clínicas, University of São Paulo Medical School (approval number CAAE 54174421.0.0000.0068). As this was a retrospective study using de-identified clinical and echocardiographic data, informed consent was waived.

### Study design

This was a retrospective, observational, single-center study conducted between January 2006 and September 2021. The study included patients diagnosed with CCC and significant ventricular dysfunction at baseline, defined as an LVEF less than 40%.

Patients were divided into two groups according to the LVEF trajectory: the positive LVRR (PRR) group, which consisted of patients who, at the initial assessment, had a transthoracic echocardiogram (TTE) showing LVEF<40% and follow-up TTE with LVEF≥40% or an absolute increase in LVEF of ≥ 10%, with a minimum interval of 6 months between TTEs; and the negative LVRR (NRR) group, which consisted of patients who, at the initial assessment, had TTE with LVEF<40% and, at follow-up, did not achieve LVEF>40% or an absolute increase in LVEF of ≥ 10%, with a minimum interval of 6 months between TTEs.

### Participants and data collection

Given the retrospective nature of the study, the sample size was determined by the availability of eligible patients within the study period. The inclusion criteria were patients who were 18 years or older, had a confirmed diagnosis of CCC on the basis of positive serology for *T. cruzi* by two distinct methods (indirect immunofluorescence, indirect hemagglutination, or enzyme-linked immunosorbent assay) [11], and had an LVEF<40%. Patients were excluded if they lacked quantitative LVEF evaluation, underwent any TTE outside the institution of the study, or had primary valvular disease, a history of valvular surgery, ischemic cardiomyopathy, or any other known concomitant cardiac etiology.

Data were collected from the electronic medical records of patients with CCC and HFrEF at the hospital. Computerized data were provided by the institutional medical information department, whereas noncomputerized clinical data were analyzed by HF specialists. For each included patient, clinical-epidemiological characteristics, as well as laboratory test results, were collected at 2 time points: (1) 3 months before or 3 months after the first TTE (Time 1–T1) and (2) 3 months before or 3 months after the second TTE (Time 2–T2). Mortality data were obtained from the São Paulo Health Department and the patients' medical record and no contact with live participants was done.

### Clinical outcomes

Patient management followed standard clinical practices as determined by the attending physicians according to local guidelines [12]. The primary outcome was a composite of all-cause mortality and heart transplantation, and secondary outcomes included isolated all-cause mortality and PRR.

## Echocardiographic measurements

The evaluation of ventricular function was performed through TTE conducted by experienced echocardiographers from the same institution, following specialty guidelines [13]. The first examination was conducted at the beginning of the follow-up period, and the second determined whether the patient would be placed in the PRR or NRR group. Simpson's method was applied when segmental wall motion abnormalities were present; otherwise, the Teichholz method was used. Key variables collected included LVEF, left ventricular end-diastolic and end-systolic diameters (LVEDD, LVESD), left atrial diameter (LAD), right ventricular dysfunction, pulmonary artery systolic pressure (PASP), and degrees of tricuspid and mitral regurgitation (TR, MR).

## Statistical analysis

To compare patients in the PRR and NRR groups across clinical-epidemiological, laboratory, and imaging data, as well as device usage, the Kolmogorov–Smirnov test was applied to assess the distribution of numerical variables, which are expressed as medians and interquartile ranges. Between group comparisons were performed via the Mann–Whitney U test. Categorical variables were compared via Pearson's chi-square test or, when expecting small values, Fisher's exact test.

Propensity score matching (PSM) was used to adjust for baseline differences between groups, matching patients at a 1:1 ratio using a comprehensive set of variables and a caliper width of 0.2 times the standard deviation of the logit. The PSM included the following data (at T1): systolic blood pressure (SBP), diastolic BP (DBP), heart rate (HR), New York Heart Association (NYHA) functional class, sex, age, comorbidities (hypertension, diabetes mellitus, dyslipidemia, atrial arrhythmias, alcohol use, smoking, chronic obstructive pulmonary disease or asthma, stroke or transient ischemic attack, hypothyroidism, and myocardial infarction), baseline LVEF, baseline LVEDD, all medications in use, the estimated glomerular filtration rate, and the use of triple therapy [14–17].

Cox proportional hazards models were applied to adjust for potential covariates and detect independent predictors of all-cause mortality or heart transplantation during long-term follow-up. Variables demonstrating a p value < 0.05 in univariate analyses that did not exhibit high collinearity (VIF > 10) or high correlation (Spearman correlation > 0.6) were then subjected to multivariate Cox analysis via the backward likelihood ratio (LR). If two variables presented a high correlation, the one with the highest beta was included in analysis. Kaplan–Meier survival analysis with log-rank tests was used to compare time-dependent outcomes. Binary logistic regression was used to identify factors associated with PRR before PSM. All analyses were conducted via IBM SPSS, version 29.0 (IBM Corp., Armonk, NY, USA), with p < 0.05 indicating statistical significance.

## Results

### Baseline clinical characteristics (T1)

Between January 2006 and September 2021, a total of 9,617 patients were serologically diagnosed with Chagas disease. After applying the exclusion criteria, 1,043 patients were evaluated, with 221 (21.2%) classified as having PRR and 822 (78.8%) as having NRR S1 Fig). When baseline (T1) characteristics before PSM were compared, significant differences were noted: the PRR group had a greater proportion of females, was older, and had a greater prevalence of hypertension and atrial arrhythmias. In terms of the ECG findings, the PRR group presented lower incidences of left anterior fascicular block and first-degree atrioventricular block, as well as a reduced density of ventricular extrasystoles on 24-hour Holter monitoring. In terms of medication use, while both groups were comparable in the use of ACE inhibitors/angiotensin receptor blockers (ACEis/ARBs), the PRR group had lower usage rates for beta-blockers (BBs) and spironolactone and was less likely to use triple therapy (ACEis/ARBs, BBs and spironolactone) than the NRR group was (S1, S2 and S3 Table). Baseline TTE parameters revealed that the PRR group had a greater median LVEF and lower median LVEDD and LVESD

values (S4 Table). After PSM, a total of 89 PRR patients and 89 NRR patients were included in the primary outcome analysis. The differences observed between the PRR and NRR groups were corrected after PSM (Table 1).

### Changes in clinical and TTE characteristics during Follow-up (T2)

In a cohort of 1,043 patients, those in the PRR group were generally older and presented higher levels of SBP and DBP, lower BNP levels, and fewer cases in NYHA classes III and IV than did those in the NRR group S5 Table) at T2. The NRR group had lower sodium levels and higher urea and creatinine levels, while both groups were similar in terms of the estimated glomerular filtration rate (S6 Table). With respect to HFrEF treatment, the prevalence of medication use was similar between the groups (Fig 1). Device usage differed, with a higher prevalence of cardiac resynchronization therapy implantation in the NRR group, whereas pacemaker implantation was more common in the PRR group (S6 Table. The PRR group demonstrated a significant median reduction in LVEDD and LVESD, alongside an improvement in LVEF. This group also had lower rates of right ventricular dysfunction, moderate-to-severe MR, and TR. Additionally, the median PASP was lower in the PRR group (S7 Table).

When the clinical characteristics at T2 were compared among the 178 individuals after PSM (Table 2), the PRR group had lower median HR and BNP levels than did the NRR group. With respect to HFrEF treatment at T2, both groups were similar in the use and doses of ACEis/ARBs, BBs, and spironolactone, as well as the use of triple therapy. However, while dose adjustments for carvedilol, spironolactone, hydralazine, nitrate, and amiodarone were observed in both groups, only the NRR group showed statistically significant changes in the doses of captopril and furosemide (S8 and S9 Table). The PRR group also exhibited superior TTE parameters at T2, with a higher median LVEF and lower LVEDD, LVESD, LAD, and PASP. Additionally, this group had a greater prevalence of normal right ventricular function and a lower prevalence of moderate-to-severe MR (Table 2, Fig 2).

### Primary and secondary outcome analysis

Over the 15-year follow-up period, 65 primary outcome events were recorded. In the PRR group, 22 (24.7%) events occurred, including 19 (21.3%) deaths and 3 (3.4%) heart transplants. In the NRR group, 43 (48.3%) events were observed, consisting of 37 (41.6%) deaths and 6 (6.7%) heart transplants (S10 Table). The mean event-free survival was 10.308 years (95% CI, 9.473–11.143) in the PRR group and 8.122 years (95% CI, 7.181–9.063) in the NRR group (p = 0.002). When all-cause mortality alone was analyzed, the PRR group continued to have better survival than the NRR group did (Fig 3).

We used a Cox regression model to identify other potential factors associated with the primary outcome (S11 Table). After excluding variables that lost significance in the bivariate analysis or exhibited multicollinearity, the following 16 variables were included in the multivariate model: PRR status, SBP (T1), age (T2), furosemide use (T1), hydralazine use (T1), hydralazine dose (T1), furosemide dose (T1), NYHA functional class (T2), SBP (T2), serum sodium level (T2), hydralazine dose (T2), LVEDD (T1), LVESD (T1), moderate-to-severe TR status (T1), and right ventricular dysfunction status (T2). The absence of data limited the analysis to 162 patients (91.0%) included after PSM. Six variables retained their prognostic significance: PRR status, NYHA functional class (T2), serum sodium quartile (T2), hydralazine use (T1), LVEDD quartile (T1), and moderate-to-severe TR status (T1) (Table 3) (S2 Fig).

### Predictors of PRR in Patients with CCC

To evaluate predictors associated with PRR status, the entire cohort of 1,043 patients was analyzed. Univariate analysis revealed several factors potentially influencing PRR (Table 4). After excluding variables with multicollinearity or high correlation indices, multivariate analysis was performed. Complete datasets were available for 557 patients (53.4%). Baseline use of triple therapy (OR 0.564; 95% CI 0.356–0.893; p = 0.0015), greater LVDD (OR 0.920; 95% CI 0.889–0.952;

**Table 1. Baseline (T1) Clinical Characteristics of 178 Patients Evaluated for Left Ventricular Reverse Remodeling (After Propensity Score Matching).**

| Characteristics | Total (n)* | All patients | PRR (n)* | PRR | NRR (n)* | NRR | P value |
|---|---|---|---|---|---|---|---|
| Age (years) | 178 | 60.0 (52.0–67.0) | 89 | 61.0 (54.0–68.0) | 89 | 58.0 (49.5–65.5) | 0.140 |
| Sex [Male (%)] | 178 | 98 (55.1) | 89 | 47 (52.8) | 89 | 51 (57.3) | 0.547 |
| NYHA Functional Class [n (%)] | 178 | | 89 | | 89 | | 0.997 |
| I | | 61 (34.3) | | 30 (33.7) | | 31 (34.8) | |
| II | | 77 (43.3) | | 39 (43.8) | | 38 (42.7) | |
| III | | 33 (18.5) | | 16 (18.0) | | 17 (19.1) | |
| IV | | 7 (3.9) | | 4 (4.5) | | 3 (3.4) | |
| Time of onset of symptoms (months) | 81 | 12.0 (3.0–42.0) | 43 | 9.0 (3.0–36.0) | 38 | 12.0 (3.8–48.0) | 0.515 |
| HR (bpm) | 178 | 70 (60–80) | 89 | 70 (60–80) | 89 | 70 (64–78) | 0.887 |
| SBP (mmHg) | 178 | 120.0 (100.0–130.0) | 89 | 120 (110–130) | 89 | 120 (100–130) | 0.426 |
| BPD (mmHg) | 178 | 80.0 (60.0–80.0) | 89 | 75 (67–87) | 89 | 80 (60–80) | 0.441 |
| Comorbidities [n (%)] | 178 | | 89 | | 89 | | |
| SAH | | 75 (42.1) | | 39 (43.8) | | 36 (40.4) | 0.649 |
| Diabetes Mellitus | | 33 (18.5) | | 18 (20.2) | | 15 (16.9) | 0.563 |
| Stroke/TIA | | 24 (13.5) | | 11 (12.4) | | 13 (14.6) | 0.661 |
| AMI | | 2 (1,1) | | 1 (1.1) | | 1 (1.1) | 1,000 |
| Atrial Tachyarrhythmias (AF, Atrial Flutter, AT) | | 66 (37.1) | | 38 (42.7) | | 28 (31.5) | 0.121 |
| Laboratory | | | | | | | |
| Hemoglobin (g/dL) | 177 | 14.1 (13.0–15.2) | 88 | 14.2 (13.0–15.3) | 89 | 14.1 (12.8–15.0) | 0.237 |
| Sodium (mEq/L) | 173 | 139.0 (138.0–141.0) | 87 | 140.0 (138.0–142.0) | 86 | 139.0 (138.0–141.0) | 0.503 |
| Potassium (mEq/L) | 173 | 4.4 (4.2–4.8) | 87 | 4.4 (4.1–4.8) | 86 | 4.4 (4.2–4.7) | 0.536 |
| eGFR (ml/min)† | 178 | 69.4 (53.8–84.7) | 89 | 69.4 (52.8–85.4) | 89 | 69.4 (56.3–84.3) | 0.588 |
| BNP (pg/mL) | 49 | 368.0 (165.0–993.0) | 27 | 541.0 (179.0–1638.0) | 22 | 226.0 (148.3–659.5) | 0.059 |
| ECG | 121 | | 58 | | 63 | | |
| RBBB | | 42 (34.7) | | 18 (31.0) | | 24 (38.1) | 0.415 |
| LBBB | | 14 (11.6) | | 6 (10.3) | | 8 (12.7) | 0.686 |
| LAFB | | 32 (26.4) | | 13 (22.4) | | 19 (30.2) | 0.335 |
| PVC | | 34 (28.1) | | 19 (32.8) | | 15 (23.8) | 0.274 |
| AVBs | | 18 (14.9) | | 6 (10.3) | | 12 (19.0) | 0.179 |
| First-Degree AVB | | 14 (11.6) | | 5 (8.6) | | 9 (14.3) | 0.330 |
| 24-hour Holter | 104 | | 57 | | 47 | | |
| PVC in 24-hours (%) | | 3.0 (0.7–6.2) | | 1.9 (0.5–6.0) | | 4.0 (0.8–6.3) | 0.167 |
| Treatment (Use) [n(%)] | | | | | | | |
| ACEI/ARB | 178 | 154 (86.5) | 89 | 77 (86.5) | 89 | 77 (86.5) | 1,000 |
| BB | 178 | 141 (79.2) | 89 | 69 (77.5) | 89 | 72 (80.9) | 0.579 |
| Spironolactone | 178 | 80 (44.9) | 89 | 38 (42.7) | 89 | 42 (47.2) | 0.547 |
| Furosemide | 178 | 100 (56.2) | 89 | 51 (57.3) | 89 | 49 (55.1) | 0.763 |
| Hydralazine | 178 | 12 (6.7) | 89 | 4 (4.5) | 89 | 8 (9.0) | 0.232 |
| Nitrate | 178 | 9 (5.1) | 89 | 3 (3.4) | 89 | 6 (6.7) | 0.496 |
| Amiodarone | 178 | 21 (11.8) | 89 | 8 (9.0) | 89 | 13 (14.6) | 0.245 |
| Triple Therapy‡ | 178 | 63 (35.4) | 89 | 31 (34.8) | 89 | 32 (36.0) | 0.875 |
| First TTE | | | | | | | |
| LVEF (%) | 178 | 30.0 (25.0–35.0) | 89 | 30.0 (25.0–37.0) | 89 | 29.0 (25.0–35.0) | 0.103 |
| LVEDD (mm) | 178 | 62.0 (58.0–67.0) | 89 | 61.0 (57.0–66.0) | 89 | 63.0 (58.0–67.0) | 0.061 |

*(Continued)*

**Table 1.** (Continued)

| Characteristics | Total (n)* | All patients | PRR (n)* | PRR | NRR (n)* | NRR | P value |
|---|---|---|---|---|---|---|---|
| LVESD (mm) | 176 | 52.0 (48.0–58.0) | 88 | 50.5 (46.3–58.0) | 88 | 54.0 (49.0–58.0) | 0.061 |
| LAD (mm) | 177 | 45.0 (41.0–49.0) | 88 | 45.0 (39.3–48.8) | 89 | 45.0 (41.0–49.0) | 0.408 |
| RV dysfunction [n (%)] | 178 | | 89 | | 89 | | 0.607 |
| Absent | | 97 (54.5) | | 45 (50.6) | | 52 (58.4) | |
| Light | | 37 (20.8) | | 22 (24.7) | | 15 (16.9) | |
| Moderate | | 36 (20.2) | | 18 (20.2) | | 18 (20.2) | |
| Serious | | 8 (4.5) | | 4 (4.5) | | 4 (4.5) | |
| Moderate or severe MR [n (%)] | 173 | 88 (50.9) | 85 | 37 (43.5) | 88 | 51 (58.0) | 0.058 |
| Moderate or severe TR [n (%)] | 165 | 55 (33.3) | 82 | 28 (34.1) | 83 | 27 (32.5) | 0.826 |
| PASP (mmHg) | 118 | 36.5 (30.0–46.0) | 60 | 35.0 (28.0–45.0) | 58 | 38.0 (30.0–48.0) | 0.250 |

Values are n (%) or median (IQR)

*N: number of patients with available data

† Calculated by the CKD-EPI formula (ml/min)

‡ Triple Therapy: ACEI/ARB, BB and Spironolactone

LVRR, left ventricular and reverse remodeling; M, male; HR, heart rate; NYHA, New York Heart Association; SBP, systolic blood pressure; DBP, diastolic blood pressure; SAH, systemic arterial hypertension; TIA, transient ischemic attack; AMI, acute myocardial infarction; AF, atrial fibrillation; AT, atrial tachycardia; eGFR, estimated glomerular filtration rate; BNP, brain natriuretic peptide; ECG, electrocardiogram; RBBB, right bundle branch block; LBBB, left bundle branch block; LAFB, left anterior fascicular block; PVC, premature ventricular contraction; AVB, atrioventricular block; ACEI, angiotensin-converting enzyme inhibitors; ARB, angiotensin receptor blockers; BB, beta blocker; TTE, transthoracic echocardiogram; LVEF, left ventricular ejection fraction; LVEDD, left ventricular end-diastolic diameter; LVESD, left ventricular end-systolic diameter; LAD, left atrium diameter; RV, right ventricle; MR, mitral regurgitation; TR, tricuspid regurgitation; PASP, pulmonary artery systolic pressure.

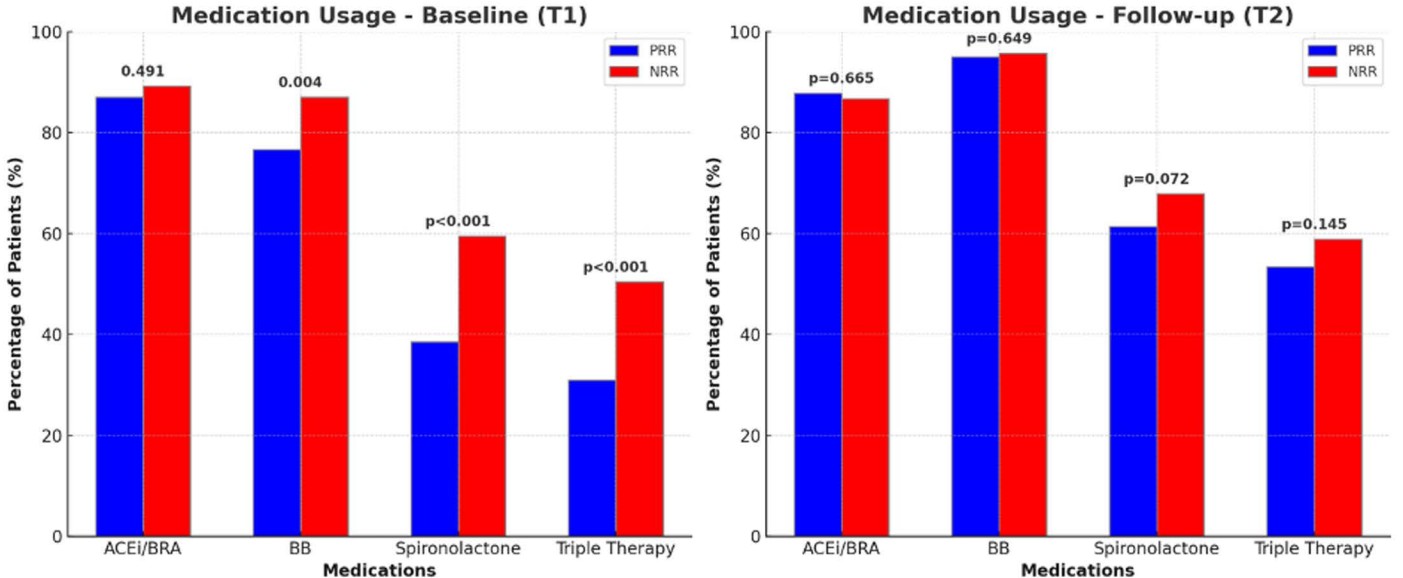

**Fig 1. Comparison of Medication Usage Between Positive (PRR) and Negative Reverse Remodeling (NRR) Groups at Baseline (T1) and Follow-up (T2).** The percentage of patients using heart failure medications at baseline (Time 1 – T1) and follow-up (Time 2 – T2) is illustrated. **(A)** Compared to the NRR group, the PRR group had significantly lower use of beta-blockers (BBs) (p = 0.004), spironolactone (p < 0.001), and triple therapy (ACEIs/ARBs, BBs, and spironolactone) (p < 0.001). **(B)** There were no significant differences. ACEis/ARBs: angiotensin-converting enzyme inhibitors/angiotensin receptor blockers.

**Table 2. Follow-up (T2) Clinical Characteristics of 178 Patients Evaluated for Left Ventricular Reverse Remodeling After Drug Optimization (After Propensity Score Matching).**

| Characteristics | Total (n)* | All patients | PRR (n)* | PRR | NRR (n)* | NRR | P value |
|---|---|---|---|---|---|---|---|
| Age (years) | 178 | 62.5 (55.0–69.3) | 89 | 64.0 (56.0–69.5) | 89 | 60.0 (51.0–69.5) | 0.177 |
| NYHA Functional Class [n (%)] | 177 | | 89 | | 88 | | 0.257 |
| I | | 61 (34.5) | | 36 (40.4) | | 25 (28.4) | |
| II | | 74 (41.8) | | 34 (38.2) | | 40 (45.5) | |
| III | | 34 (19.2) | | 14 (15.7) | | 20 (22.7) | |
| IV | | 8 (4.5) | | 5 (5.6) | | 3 (3.4) | |
| HR (bpm) | 178 | 65 (60–72) | 89 | 62 (60–70) | 89 | 68 (60–74) | 0.019 |
| SBP (mmHg) | 178 | 110 (100–120) | 89 | 120 (100–130) | 89 | 110 (98–120) | 0.188 |
| BPD (mmHg) | 178 | 70 (60–80) | 89 | 70 (60–80) | 89 | 70 (60–80) | 0.212 |
| Laboratory | | | | | | | |
| Hemoglobin (g/dL) | 178 | 13.7 (12.7–14.8) | 89 | 13.8 (12.8–15.2) | 89 | 13.4 (12.7–14.6) | 0.155 |
| Sodium (mEq/L) | 176 | 139.0 (138.0–142.0) | 88 | 139.5 (138.0–142.0) | 88 | 139.0 (137.0–142.0) | 0.384 |
| Potassium (mEq/L) | 176 | 4.5 (4.2–4.8) | 88 | 4.5 (4.2–4.8) | 88 | 4.5 (4.1–4.8) | 0.381 |
| eGFR (ml/min)† | 178 | 64.5 (48.0–79.5) | 89 | 65.6 (48.8–79.4) | 89 | 63.9 (45.7–80.7) | 0.941 |
| BNP (pg/mL) | 121 | 246.0 (134.0–791.0) | 60 | 160.0 (111.5–498.3) | 61 | 472.0 (168.5–1073.0) | 0.003 |
| Treatment (Use) [n (%)] | | | | | | | |
| ACEI/ARB | 178 | 154 (86.5) | 89 | 80 (89.9) | 89 | 74 (83.1) | 0.188 |
| BB | 178 | 167 (93.8) | 89 | 83 (93.3) | 89 | 84 (94.4) | 0.756 |
| Spironolactone | 178 | 115 (64.6) | 89 | 56 (62.9) | 89 | 59 (66.3) | 0.638 |
| Furosemide | 178 | 118 (66.3) | 89 | 57 (64.0) | 89 | 61 (68.5) | 0.526 |
| Hydralazine | 178 | 38 (21.3) | 89 | 18 (20.2) | 89 | 20 (22.5) | 0.714 |
| Nitrate | 178 | 25 (14.0) | 89 | 11 (12.4) | 89 | 14 (15.7) | 0.518 |
| Triple Therapy‡ | 178 | 96 (53.9) | 89 | 46 (51.7) | 89 | 50 (56.2) | 0.548 |
| CRT | 178 | 30 (16.9) | 89 | 12 (13.5) | 89 | 18 (20.2) | 0.230 |
| Pacemaker | 178 | 43 (24.2) | 89 | 24 (27.0) | 89 | 19 (21.3) | 0.381 |
| ICD | 178 | 28 (15.7) | 89 | 11 (12.4) | 89 | 17 (19.1) | 0.217 |
| Second TTE | | | | | | | |
| Time between the 1st and 2nd (years) | 178 | 2.0 (1.1–3.2) | 89 | 2.0 (1.1–3.0) | 89 | 2.2 (1.1–3.3) | 0.311 |
| LVEF (%) | 178 | 35.0 (29.0–42.0) | 89 | 42.0 (40.0–46.0) | 89 | 30.0 (25.0–34.0) | <0.001 |
| LVEDD (mm) | 172 | 60.0 (56.0–66.0) | 88 | 58.0 (53.3–61.8) | 84 | 64.0 (59.3–69.8) | <0.001 |
| LVESD (mm) | 171 | 50.0 (43.0. – 57.0) | 88 | 45.0 (40.0–50.0) | 83 | 55.0 (50.0–61.0) | <0.001 |
| LAD (mm) | 172 | 44.0 (40.0–50.0) | 88 | 43.0 (38.0–47.0) | 84 | 46.0 (42.0–52.0) | <0.001 |
| RV dysfunction [n (%)] | 176 | | 89 | | 87 | | <0.001 |
| Absent | | 114 (64.8) | | 66 (74.2) | | 48 (55.2) | |
| Light | | 42 (23.9) | | 20 (22.5) | | 22 (25.3) | |
| Moderate | | 14 (3.4) | | 2 (2.2) | | 12 (13.8) | |
| Important | | 6 (3.4) | | 1 (1.1) | | 5 (5.7) | |
| Moderate or severe MR [n (%)] | 178 | 79 (44.4) | 89 | 28 (31.5) | 89 | 51 (57.3) | <0.001 |
| Moderate or severe TR [n (%)] | 178 | 61 (34.3) | 89 | 27 (30.3) | 89 | 34 (38.2) | 0.269 |

*(Continued)*

**Table 2.** (Continued)

| Characteristics | Total (n)* | All patients | PRR (n)* | PRR | NRR (n)* | NRR | P value |
|---|---|---|---|---|---|---|---|
| PASP (mmHg) | 118 | 35.0 (29.0–42.3) | 54 | 30.5 (26.8–39.3) | 64 | 39.0 (30.3–54.8) | 0.001 |

Values are n (%) or median (IQR)

*N: number of patients with available data

† Calculated by the CKD-EPI formula (ml/min)

‡ Triple Therapy: ACEI/ARB, BB and Spironolactone

NYHA, New York Heart Association; HR, heart rate; SBP, systolic blood pressure; DBP, diastolic blood pressure; eGFR, estimated glomerular filtration rate; BNP, brain natriuretic peptide; ACEI, angiotensin-converting enzyme inhibitors; ARB, angiotensin receptor blockers; BB, beta blocker; CRT, cardiac resynchronization therapy; ICD, implantable cardioverter defibrillator; TTE, transthoracic echocardiogram; LVEF, left ventricular ejection fraction; LVEDD, left ventricular end-diastolic diameter; LVESD, left ventricular end-systolic diameter; LAD, left atrium diameter; RV, right ventricle; MR, mitral regurgitation; TR, tricuspid regurgitation; PASP, pulmonary artery systolic pressure.

p < 0.001), and moderate-to-severe MR (OR 0.568; 95% CI 0.361–0.896; p = 0.015) were associated with a lower likelihood of experiencing PRR (Table 4).

## Discussion

To our knowledge, this is the first study demonstrating that PRR status is significantly associated with better long-term clinical outcomes in patients with CCC and HFrEF. Additionally, we found that already using triple therapy at baseline, having a greater LVEDD, or having moderate-to-severe MR were associated with a lower chance of experiencing PRR. These findings provide critical insight into the prognostic value of PRR in this unique population, providing new data to improve CCC–HFrEF management, for which management strategies are often extrapolated from other forms of cardiomyopathy [18–22].

In contrast with a previous study on CCC–HFrEF that failed to demonstrate improved survival in PRR patients, our larger sample size and extended follow-up likely provided the statistical power needed to detect the association between PRR in patients with CCC and a 55% reduction in the risk of all-cause mortality and heart transplantation over a 15-year follow-up [10]. In line with other HF etiologies, improved LV function has been consistently linked with better clinical outcomes [9,23–26]. Beyond the improvement in LVEF, we observed significant reductions in LVEDD and LVESD, highlighting the importance of structural remodeling in predicting long-term survival [27].

We also identified some predictors of long-term mortality in HF patients with CCC and reduced LVEF (<40%) that is in line with other etiologies. In addition to PRR, which was associated with better prognosis, factors such as baseline hydralazine use, higher LVEDD at baseline, worsened NYHA functional class at follow-up (T2), presence of moderate-to-severe TR on the first TTE, and low serum sodium levels at T2 emerged as independent predictors of high mortality. The prognostic value of TR and right ventricular dysfunction is well-established in HF populations [28,29], and our findings reinforce the need for close monitoring of these patients to prognosis evaluation. Hyponatremia remains a strong indicator of poor prognosis, consistent with its known association with adverse outcomes in HFrEF [30].

The rate of PRR in our cohort was 21.2%, which falls within the range reported in the literature for other HF populations, where PRR rates vary between 9% and 40% [9,31–34]. It is well-accepted that patients with ischemic cardiomyopathy have lower rates of PRR probably due to extensive myocardial fibrosis and scarring [35,36]. In contrast, it is likely that nonischemic cardiomyopathies, including hypertensive and dilated cardiomyopathies, tend to show higher PRR rates due to less extensive fibrosis [37]. Given the pathophysiological characteristics of CCC HF, characterized by myocardial diffuse inflammation and fibrosis, the PRR rate in our study is consistent with the expectations of a lower PRR rate. The relatively modest remodeling rate reflects the unique myocardial damage caused by chronic inflammation and fibrosis in patients

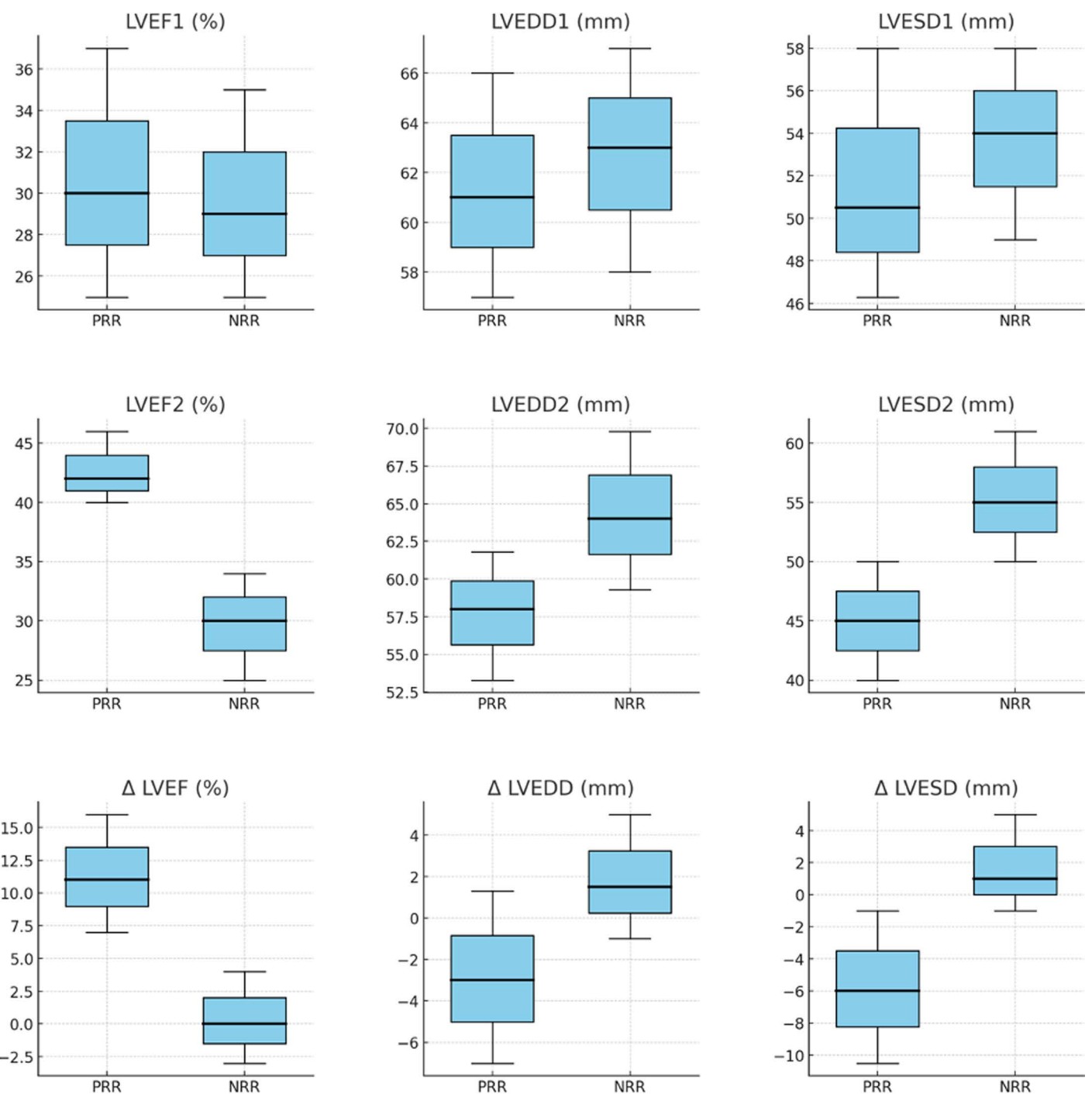

**Fig 2. Boxplots of Echocardiographic Parameters in Patients with Positive and Negative Left Ventricular Reverse Remodeling.** Measurements were evaluated at two time points: time 1 (LVEF1, LVEDD1, and LVESD1), time 2 (LVEF2, LVEDD2, and LVESD2) and the variation between the two time points (ΔLVEF, ΔLVEDD, ΔLVESD). LVEF (%), Left ventricular ejection fraction; LVEDD (mm), Left ventricular end-diastolic diameter; LVESD (mm), Left ventricular end-systolic diameter; Δ LVEF (%), Change in LVEF between time 2 and time 1; Δ LVEDD (mm), Change in LVEDD between time 2 and time 1; Δ LVESD (mm), Change in LVESD between time 2 and time 1.

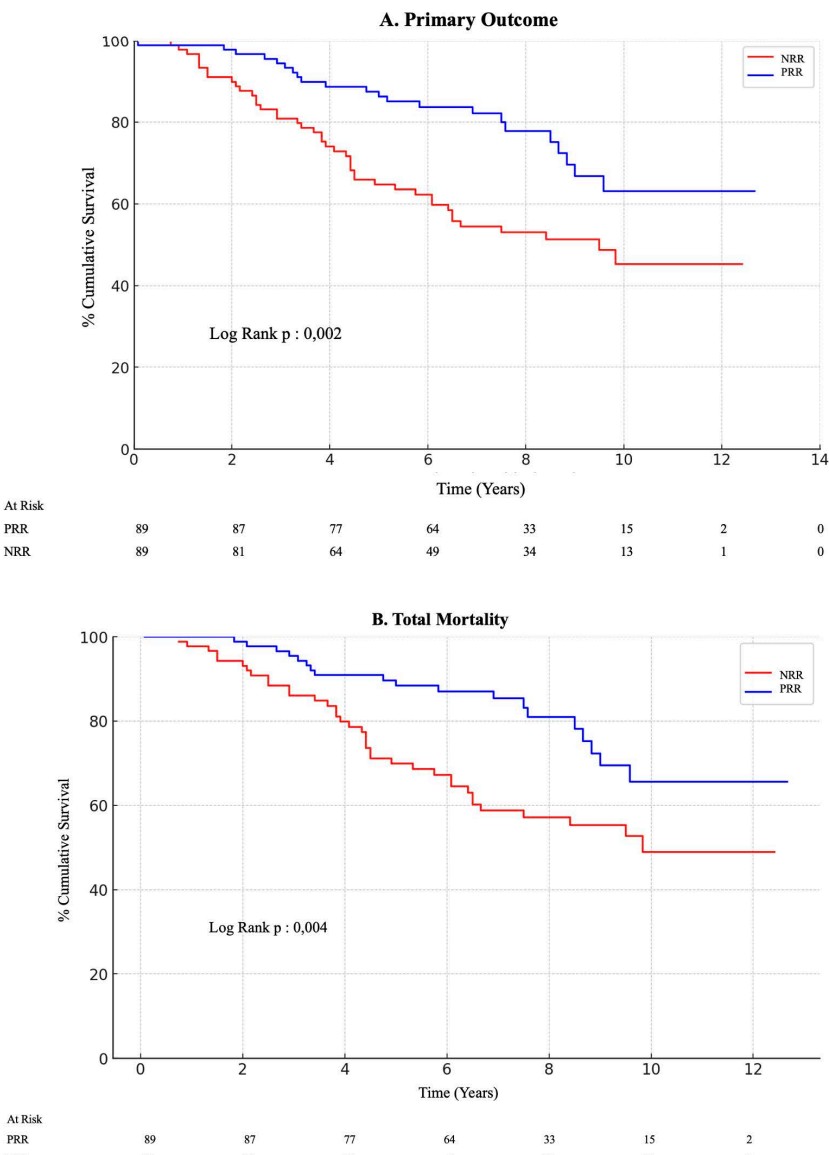

**Fig 3. Kaplan–Meier curves for the primary outcome (A) and total mortality (B).** The event-free survival time was significantly better in the positive reverse remodeling (PRR) group than in the negative reverse remodeling (NRR) group. **(A)** The mean (and 95% CI) survival times were 10.308 years (9.473–11.143) in the PRR group and 8.122 years (7.181–9.063) in the NRR group. The overall event-free survival mean for all patients was 9.290 years (8.636–9.994). **(B)** The mean (and 95% CI) survival times were 10.591 years (9.791–11.391) in the PRR group and 8.605 years (7.675–9.534) in the NRR group. The overall event-free survival mean for all patients was 9.695 years (9.060–10.329).

with CCC, which may limit the potential for full ventricular recovery. However, PRR was associated with a significant reduction in mortality, underscoring the clinical importance to plan managing CCC with HFrEF patients to obtain PRR.

An important finding in the present study is that patients with CCC who were on triple therapy at baseline were less likely to achieve PRR. This indicates that these patients, despite more aggressive treatment, may represent a group of nonresponders to triple therapy, potentially leading to poorer clinical outcomes. Early introduction of sodium–glucose cotransporter-2 (SGLT2) inhibitors or angiotensin receptor neprilysin inhibitors might benefit this subgroup [38–40]. It is important to note that this study did not collect data on prior benznidazole therapy, as its primary focus was to evaluate

**Table 3. Cox Proportional Hazards Model for Event-Free Survival (Total Mortality and Heart Transplantation)**[*]

| | Univariate Analysis | | | Multivariate Analysis | | |
|---|---|---|---|---|---|---|
| | HR | 95% CI | P value | HR | 95% CI | P value |
| PRR[†] | 0.450 | 0.269–0.753 | 0.002 | 0.472 | 0.277–0.804 | 0.006 |
| NYHA Functional Class (T2)[‡] | 1,715 | 1,293–2,275 | <0.001 | 1,621 | 1,191–2,206 | 0.002 |
| Sodium (mEq/L) (T2)[§] | 0.604 | 0.478–0.765 | <0.001 | 0.681 | 0.529–0.876 | 0.003 |
| Hydralazine use (T1)[†] | 3,696 | 1,916–7,123 | <0.001 | 3,530 | 1,760–7,080 | <0.001 |
| LVDD (T2)[§] | 1,572 | 1,237–1,999 | <0.001 | 1,345 | 1,034–1,760 | 0.027 |
| Moderate or severe TR (T1)[†] | 2,188 | 1,329–3,602 | 0.002 | 2,090 | 1,263–3,458 | 0.004 |

* The variables were analyzed after propensity score matching. Final multivariate analysis sample included 162 (91.0%) patients. Variables tested using the backward LR procedure.

†Analyzed as categorical; ‡ NYHA Functional Class analyzed in categories: I, II, III and IV.

§ Analyzed as quartiles (S12 Table).

PRR, positive reverse remodeling; T1, Time1; T2, Time2; NYHA, New York Heart Association; LVDD, left ventricular end-diastolic diameter; TR, Tricuspid regurgitation.

PRR in patients undergoing standard HFrEF therapy. Given that many patients enter specialized HF care with advanced disease, the impact of prior etiologic treatment remains uncertain. Further studies are necessary to elucidate the long-term effects of benznidazole on PRR in CCC.

Interestingly, some classical predictors of RR, such as female sex, hypertension, and the use of cardiac resynchronization therapy, did not remain significant in our multivariate analysis [24,31,41–44]. This could be explained by the unique pathophysiology of CCC, where diffuse fibrosis and chronic inflammation play a more dominant role than in other HF etiologies [45]. Furthermore, the more favorable baseline LV dimensions and lower prevalence of MR in our cohort may have influenced these results.

### Potential clinical implications

Our findings reinforce the importance of identifying patients with the potential for PRR in the management of CCC with HFrEF. Clinicians should consider strategies to optimize standard heart failure therapy, particularly in patients with predictors of PRR. Future research should explore therapeutic interventions that promote reverse remodeling in this population.

### Limitations

This study has certain limitations. First, the retrospective nature of our analysis introduces potential selection bias. However, we used PSM to create balanced groups and mitigate this bias. Second, the inclusion of patients over a 15-year period means that some of the newer HF therapies, such as SGLT2 inhibitors and angiotensin receptor neprilysin inhibitors, were underrepresented. Nevertheless, our findings remain relevant to the therapies used throughout the study period. Future studies should aim to include these novel therapies to better understand their role in promoting RR in patients with CCC. Third, the single-center design of this study may limit the generalizability of our results, underscoring the need for multicenter validation to confirm these findings. Lastly, while this study did not assess QT dispersion as a predictor of PRR, prior research has highlighted the prognostic significance of QT abnormalities in CCC, potentially linked to underlying inflammatory processes [46,47].

Future research should focus on interventions that promote PRR in patients with CCC and HFrEF. Randomized trials evaluating the role of new drug classes, such as SGLT2 inhibitors and angiotensin receptor neprilysin inhibitors, in promoting PRR and improving long-term outcomes are warranted. Additionally, understanding the molecular mechanisms that drive PRR, particularly fibrosis and inflammation, could uncover new therapeutic targets. Identifying patients most likely

**Table 4. Binary logistic regression for independent predictors of positive left ventricular reverse remodeling.**

| | Univariate analysis | | | | Multivariate analysis | | | |
|---|---|---|---|---|---|---|---|---|
| | Beta | OR | 95% CI | P value | Beta | OR | 95% CI | P value |
| Sex (female | 0.412 | 1.510 | 1,121–2,035 | 0.007 | | | | |
| Age (years) | 0.022 | 1.023 | 1,009–1.036 | <0.001 | | | | |
| LVEF (every 1%) | 0.058 | 1.387 | 1,207–1,594 | <0.001 | | | | |
| LVDD (every 1mm) | -0.099 | 0.905 | 0.885–0.927 | <0.001 | -0.084 | 0.920 | 0.889–0.952 | <0.001 |
| LVSD (every 1 mm) | -0.092 | 0.912 | 0.893–0.932 | <0.001 | | | | |
| LAD (every 1mm) | -0.033 | 0.967 | 0.944–0.991 | 0.006 | | | | |
| LV septum (every 1 mm) | 0.180 | 1.197 | 1,072–1,337 | 0.001 | | | | |
| LV PP (every 1mm) | 0.177 | 1.193 | 1,056–1,349 | 0.005 | | | | |
| PASP (every 1 mmHg) | -0.020 | 0.980 | 0.963–0.997 | 0.024 | | | | |
| Moderate or severe MR | -0.526 | 0.591 | 0.432–0.808 | <0.001 | -0.565 | 0.568 | 0.361–0.896 | 0.015 |
| % PVC on Holter | -0.041 | 0.960 | 0.925–0.996 | 0.032 | | | | |
| SAH | 0.322 | 1,379 | 1,020–1,865 | 0.037 | | | | |
| SBP (every 1 mmHg) | 0.023 | 1.023 | 1,012–1.034 | <0.001 | | | | |
| DBP (every 1 mmHg) | 0.035 | 1.036 | 1,018–1,054 | <0.001 | | | | |
| HR (every 1 bpm) | 0.031 | 1.031 | 1,016–1,047 | <0.001 | | | | |
| Pacemaker use | 0.350 | 1,419 | 1,030–1,956 | 0.032 | | | | |
| CRT use | -0.679 | 0.507 | 0.312–0.823 | 0.006 | | | | |
| Use of furosemide | -0.595 | 0.539 | 0.360–0.807 | 0.003 | | | | |
| Use of BB | -0.669 | 0.489 | 0.297–0.805 | 0.005 | | | | |
| Use of spironolactone | -0.860 | 0.427 | 0.283–0.643 | <0.001 | | | | |
| Use of amiodarone | -0.728 | 0.483 | 0.277–0.841 | 0.010 | | | | |
| Use of triple therapy* | -0.823 | 0.439 | 0.287–0.671 | <0.001 | -0.574 | 0.564 | 0.356–0.893 | 0.015 |
| Presence of LAFB | -0.478 | 0.620 | 0.405–0.905 | 0.028 | | | | |
| AF/Atrial flutter/AT (ECG) | 0.697 | 2.009 | 1,101–3,666 | 0.023 | | | | |
| Presence of AVB 1st | -0.868 | 0.420 | 0.232–0.760 | 0.004 | | | | |

Because of some missing values for some patients, the final sample for multivariate analysis included 557 (53.4%) of the 1043 patients. All factors included in this analysis were from time 1 (T1).

*Triple therapy: angiotensin-converting enzyme inhibitors or angiotensin receptor blockers or angiotensin receptor neprilysin inhibitors + BB + spironolactone.

+Analyzed as categorical variables: female sex, moderate MI, SAH, use of pacemaker, use of cardiac resynchronizer, use of furosemide, use of BB, use of spironolactone, use of amiodarone, use of triple therapy, presence of BDAS (ECG), presence of AF/atrial flutter/AT (ECG), presence of AV Block 1st (ECG).

LVEF, left ventricular ejection fraction; LVDD, left ventricular end-diastolic diameter; LVSD, left ventricular end-systolic diameter; LAD, left atrial diameter; LV, left ventricle; PP, posterior wall; PASP, pulmonary artery systolic pressure; MR, mitral regurgitation; PVC, premature ventricular contraction; SAH, systemic arterial hypertension; SBP, systolic blood pressure DBP, diastolic blood pressure; HR, heart rate; BB, beta-blocker; LAFB, left anterior fascicular block; ECG, electrocardiogram; AF, atrial fibrillation; AT, atrial tachycardia; AVB, atrioventricular block.

to benefit from aggressive therapies will enable more tailored treatment strategies, improving survival in this challenging population.

## Conclusion

This study reveals that PRR is significantly associated with decreased all-cause mortality and heart transplantation rates over a 15-year follow-up period in patients with CCC and HFrEF. Baseline left ventricular dimensions, mitral regurgitation status, and the use of triple therapy were identified as key predictors of PRR, offer valuable insights for refining treatment strategies. These findings underscore the importance of early detection and aggressive intervention to promote PRR and improve clinical outcomes, suggesting a future focus on therapies that enhance ventricular recovery to reduce morbidity and mortality in this high-risk group.

## Supporting information

**S1 Table. Clinical characteristics and comorbidities of the 1043 patients analyzed for LV reverse remodeling at baseline.**
(PDF)

**S2 Table. Laboratory tests, electrocardiogram, and holter monitoring results at baseline.**
(PDF)

**S3 Table. Treatment at baseline.**
(PDF)

**S4 Table. Comparison of the first transthoracic echocardiogram results at baseline.**
(PDF)

**S5 Table. Clinical and laboratory characteristics at follow-up.**
(PDF)

**S6 Table. Treatment at follow-up.**
(PDF)

**S7 Table. Comparison of the second transthoracic echocardiogram results at follow-up.**
(PDF)

**S8 Table. Medication doses at T1 and T2 in the PRR group.**
(PDF)

**S9 Table. Medication doses at T1 and T2 in the NRR group.**
(PDF)\

**S10 Table. Incidence of primary and secondary outcomes during follow-up.**
(PDF)

**S11 Table. Univariate Cox proportional risk analysis for event-free survival.**
(PDF)

**S12 Table. Quartiles of numeric variables with potential impacts on event-free survival.**
(PDF)

**S1 Fig. Flow diagram of patient selection and analysis for left ventricular reverse remodeling.** *Systemic arterial hypertension, diabetes mellitus; dyslipidemia; atrial arrhythmias, alcohol use, smoking, chronic obstructive pulmonary

disease or asthma, stroke or transient ischemic attack, hypothyroidism, acute myocardial infarction. ** Angiotensin-converting enzyme inhibitors/angiotensin receptor blockers/neprilysin and angiotensin receptor inhibitors, beta-blockers, spironolactone, furosemide, thiazide, hydralazine, nitrate, digoxin, and amiodarone. TTE, transthoracic echocardiography; LVEF, left ventricular ejection fraction; CCC, chronic Chagasic cardiomyopathy; PRR, positive left ventricular reverse remodeling; NRR, negative left ventricular reverse remodeling; SBP, systolic blood pressure; DBP, diastolic blood pressure; HR, heart rate; FC, functional class; NYHA, New York Heart Association; LVDD, left ventricular end-diastolic diameter; eGFR, estimated glomerular filtration rate.
(TIF)

**S2 Fig. Predictors of mortality in patients with Chagasic cardiomyopathy after multivariate analysis.** (A) Presence of PRR (B) Functional Class (NYHA) at T2. (C) Serum sodium levels at T2. (D) Use of hydralazine at T1. (E) LVEDD in the first transthoracic echocardiogram (TTE). (F) Presence of moderate or severe tricuspid regurgitation in the first TTE. PRR, positive reverse remodeling; LVRR-, negative left ventricular reverse remodeling; FC2, New York Heart Association Functional Class at moment 2; LVEDD, left ventricular end-diastolic diameter; TR, tricuspid regurgitation; Q1, 1st quartile; Q2, 2nd quartile; Q3, 3rd quartile; Q4, 4th quartile.
(TIF)

## Author contributions

**Conceptualization:** Maria Tereza Sampaio de Sousa Lira, Silas Ramos Furquim, Daniel Catto de Marchi, Pamela Camara Maciel, Rafael Cavalcanti Tourinho Dantas, Bruno Biselli, Paulo Roberto Chizzola, Robinson Tadeu Munhoz, Felix José Alvarez Ramires, Barbara Maria Ianni, Fábio Fernandes, Silvia Moreira Ayub-Ferreira, Edimar Alcides Bocchi.

**Data curation:** Maria Tereza Sampaio de Sousa Lira, Silas Ramos Furquim, Daniel Catto de Marchi, Pamela Camara Maciel, Robinson Tadeu Munhoz, Felix José Alvarez Ramires, Barbara Maria Ianni, Fábio Fernandes.

**Formal analysis:** Maria Tereza Sampaio de Sousa Lira, Fábio Fernandes, Edimar Alcides Bocchi.

**Investigation:** Maria Tereza Sampaio de Sousa Lira, Rafael Cavalcanti Tourinho Dantas, Silvia Moreira Ayub-Ferreira, Edimar Alcides Bocchi.

**Methodology:** Maria Tereza Sampaio de Sousa Lira, Silas Ramos Furquim, Pamela Camara Maciel, Rafael Cavalcanti Tourinho Dantas, Bruno Biselli, Paulo Roberto Chizzola, Silvia Moreira Ayub-Ferreira, Eduardo Gomes Lima, Edimar Alcides Bocchi.

**Project administration:** Maria Tereza Sampaio de Sousa Lira, Silas Ramos Furquim.

**Resources:** Maria Tereza Sampaio de Sousa Lira.

**Supervision:** Fábio Fernandes, Silvia Moreira Ayub-Ferreira, Eduardo Gomes Lima, Edimar Alcides Bocchi.

**Validation:** Felix José Alvarez Ramires, Barbara Maria Ianni, Eduardo Gomes Lima.

**Visualization:** Maria Tereza Sampaio de Sousa Lira, Silvia Moreira Ayub-Ferreira, Eduardo Gomes Lima, Edimar Alcides Bocchi.

**Writing – original draft:** Maria Tereza Sampaio de Sousa Lira, Silas Ramos Furquim, Daniel Catto de Marchi, Rafael Cavalcanti Tourinho Dantas, Bruno Biselli, Fábio Fernandes, Eduardo Gomes Lima, Edimar Alcides Bocchi.

**Writing – review & editing:** Maria Tereza Sampaio de Sousa Lira, Rafael Cavalcanti Tourinho Dantas, Eduardo Gomes Lima, Edimar Alcides Bocchi.

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
