## [Decision Letter · Decision Letter 0]

26 Feb 2025

PNTD-D-25-00077Left ventricular reverse remodeling: a predictor of survival in chagasic cardiomyopathy patients with a reduced ejection fractionPLOS Neglected Tropical Diseases Dear Dr. Lira, Thank you for submitting your manuscript to PLOS Neglected Tropical Diseases. After careful consideration, we feel that it has merit but does not fully meet PLOS Neglected Tropical Diseases's publication criteria as it currently stands. Therefore, we invite you to submit a revised version of the manuscript that addresses the points raised during the review process. Please submit your revised manuscript within 30 days Apr 27 2025 11:59PM. If you will need more time than this to complete your revisions, please reply to this message or contact the journal office at plosntds@plos.org. Please include the following items when submitting your revised manuscript:* A rebuttal letter that responds to each point raised by the editor and reviewer(s). You should upload this letter as a separate file labeled 'Response to Reviewers '. This file does not need to include responses to any formatting updates and technical items listed in the 'Journal Requirements' section below.* A marked-up copy of your manuscript that highlights changes made to the original version. You should upload this as a separate file labeled 'Revised Manuscript with Track Changes '.* An unmarked version of your revised paper without tracked changes. You should upload this as a separate file labeled 'Manuscript '. If you would like to make changes to your financial disclosure, competing interests statement, or data availability statement, please make these updates within the submission form at the time of resubmission. Guidelines for resubmitting your figure files are available below the reviewer comments at the end of this letter. We look forward to receiving your revised manuscript. Kind regards, Claudia Ida BrodskynSection EditorPLOS Neglected Tropical Diseases Claudia BrodskynSection EditorPLOS Neglected Tropical Diseases

Shaden Kamhawi

co-Editor-in-Chief

Paul Brindley

co-Editor-in-Chief

 **Additional Editor Comments:** The manuscript Left ventricular reverse remodeling: a predictor of survival in chagasic cardiomyopathy patients with a reduced ejection fraction is very important for the field and the reviewer have pointed out some questions that deserve attention in order to improve the quality of the manuscript.  **Journal Requirements:**

At this stage, the following Authors/Authors require contributions: Maria Tereza Sampaio de Sousa Lira, Silas Ramos Furquim, Daniel Catto de Marchi, Pamela Camara Maciel, Rafael Cavalcanti Tourinho Dantas, Bruno Biselli, Paulo Roberto Chizzola, Robinson Tadeu Munhoz, Felix Jose Alvarez Ramires, Barbara Maria Ianni, Fábio Fernandes, Silvia Moreira Ayub-Ferreira, Eduardo Gomes Lima, and Edimar Alcides Bocchi. Please ensure that the full contributions of each author are acknowledged in the "Add/Edit/Remove Authors" section of our submission form.

5) In the online submission form, you indicated that "data will be made available upon request". All PLOS journals now require all data underlying the findings described in their manuscript to be freely available to other researchers, either

- In a public repository

- Within the manuscript itself

- Uploaded as supplementary information.

 **Reviewers' comments:** Reviewer's Responses to Questions

**Key Review Criteria Required for Acceptance?**

**Methods** :

-Are the objectives of the study clearly articulated with a clear testable hypothesis stated?

-Is the study design appropriate to address the stated objectives?

-Is the population clearly described and appropriate for the hypothesis being tested?

-Is the sample size sufficient to ensure adequate power to address the hypothesis being tested?

-Were correct statistical analysis used to support conclusions?

-Are there concerns about ethical or regulatory requirements being met?

Reviewer #1: see comments

**Results**

-Does the analysis presented match the analysis plan?

-Are the results clearly and completely presented?

-Are the figures (Tables, Images) of sufficient quality for clarity?

Reviewer #1: see comments

**Conclusions**

-Are the conclusions supported by the data presented?

-Are the limitations of analysis clearly described?

-Do the authors discuss how these data can be helpful to advance our understanding of the topic under study?

-Is public health relevance addressed?

Reviewer #1: see comments

**Editorial and Data Presentation Modifications?**

Reviewer #1: see comments

**Summary and General Comments**

Reviewer #1: PNTD-D-25-00077

General comments:

The MS intitled “Left ventricular reverse remodeling: a predictor of survival in chagasic cardiomyopathy patients with a reduced ejection fraction” by Lira et al. uses a retrospective and observational strategy to study whether LVRR is associated with improved survival in a Brazilian cohort of patients with Chagas heart disease and LVEF<40%. A previous study by Nakazone et al. 2022 analyzed another Brazilian cohort (156 cardiac Chagas disease patients in a 10-years period) showing that LVRR had no impact on mortality. Therefore, although it is not an original analysis, the results presented in the study by Lira et al. are quite interesting: it does not corroborate previous study by Nakazone, rather shows that PRR was associated with reduced primary outcome and increased survival, suggesting focus on therapy that promotes LVRR in CCC patients. In this MS ethical issues were considered, and statistical analysis seems appropriate. Thus, this MS may be considered for publication after clarification of some points.

Major Comments:

Studies by Laranja et al. 1956 (parasite), Higuchi et al. 1993, Reis et al. 1993 (parasite antigens) and several others showing parasite DNA in heart tissue of Chagas disease patients pointed to parasite persistence in the heart tissue in chronic Chagas cardiac disease. Considering the etiopathogenic basis of chronic Chagas heart disease and potential therapeutic targets, Bocchi et al. 2017 (Figure 2) enrolled inflammation (IFN gamma-enriched and innate immunity), partial parasite control and “persistent low-grade parasitism” (direct parasite damage). Further, they pointed out benznidazole treatment as a specific predictor of prognosis besides others such as EKG abnormalities (as QT dispersion, prolonged filtered QRS duration), midrange LVEF concomitant with complex VPCs, and others. Compared with no treatment, benznidazole treatment was associated with reduced progression of Chagas disease (Viotti et al. 2006). Recently, Hasslocher et al. 2020 showed, in a long-term follow-up (more than 30-y), that benznidazole treatment was associated with a decreased incidence of Chagas disease progression from indeterminate chronic form to the cardiac form and with a decreased risk of cardiovascular events. A recent cross-sectional study using a reduced number of patients suggested that patients that received benznidazole treatment years prior to administration of ACEi/ARA show reduced TNF serum levels and improved LVEF in patients with HFrEF (da Silva et al. 2024). This is quite relevant considering the discussed “… the clinical importance to plan managing CCC with HFrEF patients to obtain PRR”. Unfortunately, no data on etiological treatment are shown in the present MS (or in the previous study by Nakazone et al.), a crucial point since Chagas heart disease is a consequence of a parasitic infection, and etiological treatment should be available to eligible patients according to Brazilian Consensus and Protocols. Altogether, these issues pointed to the questions: what are the frequencies of patients treated with Benznidazole between PRR and NRR (whole groups - 221/822- and final subgroups - 89/89?). Does Benznidazole therapy impact LVRR? Please include data on etiological treatment in the Tables. What is the impact of the variable “BZN administration” on the interpretation of the data? If the etiological treatment was not administered to this cohort of 1,043 patients with Chagas disease, please refer to it in the text, preferably with a reasonable explanation for this.

Lines 106-107: “chronic chagasic cardiomyopathy (CCC) is characterized by a persistent cardiac inflammatory process and the development of fibrosis”. I agree with this, however in this equation one should include “the persistence of the parasite”. Please, rephrase this sentence. In this case, reference #2 fits well.

Lines 152-153: “determined by the attending physicians according to local guidelines.” What are/were these “local guidelines”? It is a hermetic information, please add references to enlighten the readers.

Lines 200-203: “In terms of the ECG findings, the PRR group presented lower incidences of left anterior fascicular block and first-degree atrioventricular block, as well as a reduced density of ventricular extrasystoles on 24-hour Holter monitoring.” Considering the relevance of the QT interval dispersion for prognosis in Chagas disease (Salles et al. 2003 and others) and the inflammatory component of Chagas disease as potential contributor to QT abnormalities (Cunha-Neto and Chevillard 2014, Lazzerini et al. 2015, Lazzerini et al. 2023), is there any association of the QT dispersion (or other EKG parameter) with PRR or NRR in this CCC cohort?

The final data on predictors (with 557 patients – 53.4% of the cohort shown in Table S13) is quite interesting and support the authors’ proposal and should be added to the text (as Table 4).

“Potential clinical implications” and “Limitation” sections should be added to the MS.

Minor comments:

A grammar and spelling review is required.

Line 104: According to the rules of species nomenclature “(T. cruzi)” should be omitted.

Line 139: “ETT” do you mean TTE?

PLOS authors have the option to publish the peer review history of their article (what does this mean? ). If published, this will include your full peer review and any attached files.

**Do you want your identity to be public for this peer review?** For information about this choice, including consent withdrawal, please see our Privacy Policy .

Reviewer #1: No

---

## [Decision Letter · Decision Letter 1]

10 Apr 2025

Dear MD Lira,

We are pleased to inform you that your manuscript 'Left ventricular reverse remodeling: a predictor of survival in chagasic cardiomyopathy patients with a reduced ejection fraction' has been provisionally accepted for publication in PLOS Neglected Tropical Diseases.

Best regards,

Claudia Ida Brodskyn

Section Editor

Claudia Brodskyn

Section Editor

Shaden Kamhawi

co-Editor-in-Chief

Paul Brindley

co-Editor-in-Chief

Reviewer's Responses to Questions

**Key Review Criteria Required for Acceptance?**

**Methods**

-Are the objectives of the study clearly articulated with a clear testable hypothesis stated?

-Is the study design appropriate to address the stated objectives?

-Is the population clearly described and appropriate for the hypothesis being tested?

-Is the sample size sufficient to ensure adequate power to address the hypothesis being tested?

-Were correct statistical analysis used to support conclusions?

-Are there concerns about ethical or regulatory requirements being met?

Reviewer #1: Clearly presented.

**Results**

-Does the analysis presented match the analysis plan?

-Are the results clearly and completely presented?

-Are the figures (Tables, Images) of sufficient quality for clarity?

Reviewer #1: Improved. Suggestions accepted.

**Conclusions**

-Are the conclusions supported by the data presented?

-Are the limitations of analysis clearly described?

-Do the authors discuss how these data can be helpful to advance our understanding of the topic under study?

-Is public health relevance addressed?

Reviewer #1: Conclusions are supported by the results.

**Editorial and Data Presentation Modifications?**

Reviewer #1: None.

**Summary and General Comments**

Reviewer #1: This reviewer has revised the R1 version of the manuscript carefully. Most of the criticisms and suggestions were well received and incorporated into the text. Therefore, I consider that the article can be accepted in its current format.

PLOS authors have the option to publish the peer review history of their article (what does this mean? ). If published, this will include your full peer review and any attached files.

**Do you want your identity to be public for this peer review?** For information about this choice, including consent withdrawal, please see our Privacy Policy .

Reviewer #1: **Yes: ** Joseli Lannes-Vieira

---

## [Editor Report · Acceptance letter]

Dear MD Lira,

We are delighted to inform you that your manuscript, "Left ventricular reverse remodeling: a predictor of survival in chagasic cardiomyopathy patients with a reduced ejection fraction," has been formally accepted for publication in PLOS Neglected Tropical Diseases.

Best regards,

Shaden Kamhawi

co-Editor-in-Chief

Paul Brindley

co-Editor-in-Chief
